# A Gnotobiotic Model to Examine Plant and Microbiome Contributions to Survival under Arsenic Stress

**DOI:** 10.3390/microorganisms9010045

**Published:** 2020-12-26

**Authors:** María del Carmen Molina, James F. White, Sara García-Salgado, M. Ángeles Quijano, Natalia González-Benítez

**Affiliations:** 1Área de Biodiversidad y Conservación, Departamento de Biología, Geología, Física y Química Inorgánica, Universidad Rey Juan Carlos, 28933 Móstoles, Spain; natalia.gonzalez@urjc.es; 2Department of Plant Biology, Rutgers University, New Brunswick, NJ 08901-8520, USA; white@rci.rutgers.edu; 3Departamento de Ingeniería Civil: Hidráulica y Ordenación del Territorio, Escuela Técnica Superior de Ingeniería Civil, Universidad Politécnica de Madrid, 28014 Madrid, Spain; sara.garcia@upm.es (S.G.-S.); marian.quijano@upm.es (M.Á.Q.)

**Keywords:** gnotobiotic, biome, endophytic bacteria, arsenic, healthy plant

## Abstract

So far, the relative importance of the plant and its microbiome in the development of early stages of plant seedling growth under arsenic stress has not been studied. To test the role of endophytic bacteria in increasing plant success under arsenic stress, gnotobiotic seeds of *J. montana* were inoculated with two endophytic bacteria: *Pantoea conspicua* MC-K1 (PGPB and As resistant bacteria) and *Arthrobacter* sp. MC-D3A (non-helper and non-As resistant bacteria) and an endobacteria mixture. In holobiotic seedlings (with seed-vectored microbes intact), neither the capacity of germination nor development of roots and lateral hairs was affected at 125 μM As(V). However, in gnotobiotic seedlings, the plants are negatively impacted by absence of a microbiome and presence of arsenic, resulting in reduced growth of roots and root hairs. The inoculation of a single PGPB (*P. conspicua*-MCK1) shows a tendency to the recovery of the plant, both in arsenic enriched and arsenic-free media, while the inoculation with *Arthrobacter* sp. does not help in the recovery of the plants. Inoculation with a bacterial mixture allows recovery of plants in arsenic free media; however, plants did not recover under arsenic stress, probably because of a bacterial interaction in the mixture.

## 1. Introduction

Arsenic is a natural metalloid of the earth’s crust, often with an anthropogenic origin: insecticides, mining and smelting, pesticides or fertilizers, industrial processes, coal combustion, etc. [1]. It is considered to be a non-essential metalloid for plants and animals [2]. However, it can be accumulated in plants to toxic levels with important pernicious effects, altering physiological processes, growth and modifying their morphology [3]. The conversion of As(V) to As(III) inside plants can generates free reactive oxygen species (ROS), such as superoxide radicals (O^2−^), hydroxyl radicals (OH^−^), and hydrogen peroxide (H_2_O_2_). ROS can cause unrepairable damage to important macromolecules, including lipids, proteins, carbohydrates, and DNA [4,5]. Furthermore, arsenic provokes pronounced reductions in gas exchange attributes (photosynthesis, stomatal conductance, transpiration rate, and intercellular CO_2_) and a significant reduction in chlorophyll content [6]. As a consequence of these physiological alterations, important morphological effects are observed, such as reduced leaf numbers, reduced leaf area, chlorotic appearance of leaves, reduced stem diameter, reduced dry weight, etc. [3,6,7,8]. Some plants have developed several strategies to survive this metalloid, for instance through chelation processes, transformation, accumulation in membranous organelles, extrusion [9,10,11] or translocation by xylem to shoot tissues [12]. Other plant strategies have shown the reduction of As influx by suppressing phosphate/arsenate uptake systems and/or increasing the antioxidants against ROS produced in response to arsenic [13]. However, more recently, the role of the microbiomes of plants are also being studied since many bacteria have the ability to respire and metabolize As. Bacteria from soil, rhizosphere and endophytic bacteria are being used in applications to increase crop productivity and decontamination of soil [14]. In the same way, studies have suggested that plant-associated microbes have an exceptional ability to reduce contaminant phytotoxicity by immobilizing them in roots, or through binding, accumulation, transformation or dilution in the host plant [15,16]. In particular, endophytic bacteria are able to assist plants in As(V) reduction and As(III) oxidation favoring the extrusion of As(III) to the soil or the translocation to the leaves [17,18,19], decreasing the translocation to the grain [20] and activating detoxification mechanisms [21,22]. Thus, microbes are able to transform As(III) to non-volatile organic forms, such as monomethylarsonic acid (MMA), dimethylarsinic acid (DMA) and trimethylarsine oxide (TMAO), which could be excreted to the soil [23,24] or volatile forms, such as trimethylarsine (TMA), which are easily removed by diffusion [10,25,26]. On the other hand, As can be vacuolated within the bacteria and even some bacteria can substitute arsenic for phosphorus to sustain growth [27]. Some bacteria can reduce oxidative stress and show down-regulation of Si and P transporters that eventually favor the entry of As [28]. Endophytic actinobacteria are capable of producing siderophores that bind arsenic, as a mechanism for detoxification [29]. It is possible that some of these metabolic capacities may have been acquired by horizontal transfer [30].

Inoculated plants with these bacteria or bacterial consortia improve arsenic sequestration efficiency with an evident advantage in phytoremediation [21,22,31,32]. However, there are open questions from an ecophysiological point of view. What is the role of the microbiome in plant resistance to As in early stages of seedling growth? Does remediation depend on the microbiome and all the interspecific interactions that are established between microbiome, plant and inoculum [33,34]?

*Jasione montana* L. is a biennial, rarely annual, species in family Campanulaceae. It grows on heaths and moors at high elevations in rocky districts, coastal and cliffs, quarries and natural escarpments, where the soil is thin and acid [35]. It is widely distributed and highly polymorphic with many ecological variants [36]. It can complete its development and reproductive cycle in environments that highly contaminated by arsenic [37,38], behaving as a tolerant plant [39,40]. Like all macroorganisms, *J. montana* can be considered as a complex multi-genomic organism (plant-metaorganism or plant-holobiont) constituted by the plant and its associated microbiota [41,42]. According to previous works [43], the use of gnotobiotic model plants can provide the knowledge of relative contributions of each biological component of the metaorganism. The aim of the present work is to develop an understanding of the role of the plant microbiome in plant resistance to As stress in early stages of plant growth. In order to meet this goal, we use a partial gnotobiotic functional (without streptomycin-sensitive biome) of *J. montana* to evaluate the physiological response of *J. montana* under As stress conditions and to analyze the relative roles of the plant and the associated microbiome in resistance to As stress. 

## 2. Materials and Methods 

### 2.1. Seeds Selection and Plants Collection

*J. montana* seeds were obtained from two different locations. To obtain a gnotobiotic model *of J. montana*, seeds were collected from Saint-Georges, Cantal (France) and provided by germoplasme bank (Muséum National d’Histoire Naturelle, Paris). These seeds were collected in 2010 from plants grown on acid soil without arsenic contamination. To obtain PPGB endophytic bacteria resistant to As, *J. montana* adult plants collected in 2016 from an arsenic mine in Bustarviejo (Madrid, Spain), in sandy and acid soil, with high arsenic concentrations (0.3–30 g·Kg^−1^) [38,39] were collected. After collecting, these plants were washed to remove the remains of soil, dust and other organic remains. Then, they were dried and stored in the refrigerator, which is used for the isolation of endophytic bacteria. 

### 2.2. Screening of Arsenic Effect on the Germination and Development of Seedlings 

*J. montana* seeds were washed with 2% (*v*/*v*) sodium hypochlorite for 5 min with slow rotation in a sterilized 50-mL conical centrifuge tube and germinated in 0.7% agarose (low melting point, Sigma-Aldrich) plates [44]. Germinated seeds (average of 20 per each plate) were grown for 5 days at 25 °C with increasing concentrations of As(V) from 0, 125, 250, 500, 1000, 1500, 2000 µM (HAsNa_2_O_4_·7H_2_O, Sigma-Aldrich). Two replicates at each concentration were prepared to estimate the percentages of germination, plants with developed roots (more than 5 mm) and plants whose roots developed lateral hairs (at least ten lateral hairs). During seed germination, seeds were discarded if a halo of microbes appeared around the seeds being considered such as microbial contamination. 

Anatomical and oxidative deterioration in cotyledons and incipient root tumors were also tested in seedling plants germinated in non-arsenic 0.7% agarose (control). Each plate (x 3) contained 10 seeds that were irrigated after 10 days with water or with water supplemented with 1 mM As(V) for two days. 

### 2.3. Seed Sterilization Effects on Plant Development 

To test the role of the microbiome in *J. montana* under arsenic stress, 4 treatments were made. In the control treatment (T1), seeds were not treated with antibiotics to retain holobiotic seedlings (with seed-vectored microbes intact). To obtain gnotobiotic seedings, endophytic bacteria were eliminated by several sterilization treatments. Treatment 2 (T2) seeds were washed with streptomycin sulphate (100 g l-1, Sigma-Aldrich) 24 h before germination and rinsed with miliQ water (three times) to remove the antibiotic (Verma et al. 2017). To produce a severe sterilization process, seeds were placed in germination media (0.7% agarose) with 5 µL of streptomycin (100 g L^−1^) for the duration of the experiments (10 days; T3). Treatment 4 (T4) was the combination of T2 and T3. Each treatment was developed under As stress (125 µM As[V]) and no As stress. In addition, to test the role of epiphytic microbiota, each treatment was developed with and without seeds, being washed in 2% (*v*/*v*) sodium hypochlorite for 5 min with slow rotation and rinsed three times with miliQ water to remove the sodium hypochlorite solution. A total of 16 treatments; antibiotic effect (×4), arsenic effect (×2) and hypochlorite effect (×2). Each treatment was done in triplicate, and 20 seeds were placed on each plate (Figure 1). 

### 2.4. Microscopic Microbiome Detection

A colorimetric test using 3, 3, diaminobenzidine (DAB) stain allows to detection of H_2_O_2_ (reddish-brown coloration in tissues) as a consequence of the detoxification of free reactive oxygen species (ROS) activated by superoxide dismutase in plant tissue [45]. In our experiments, use of DAB allowed visualization of reactive oxygen associated with bacteria in/on seeding roots [46]. Because plant cells secrete reactive oxygen onto bacteria that penetrate plant cells and come into contact with the root cell plasma membranes, intracellular penetration of root cells may be indicated by dark red or brown staining (H_2_O_2_) over or within root hairs and/or parenchyma cells. To increase capacity to visualize bacteria, we used aniline blue (0.01%, aqueous) as a counterstain. Slides were examined using bright field microscopy on a Zeiss Axioskope with a Spot InsightTM 4 megapixel digital camera. 

### 2.5. Isolation of Endophytic Bacteria Resistant to As(V)

Endophytic bacteria were isolated from the stems of *J. montana* collected on arsenic contaminated soil [38]. To remove bacterial and fungal epiphytes, stems were treated with 2% (*v*/*v*) sodium hypochlorite for 20 min with slow rotation and rinsed three times with milliQ water to remove the sodium hypochlorite solution. Stems were cut in in fragments (5 cm) placed on agar plates (LB-YES-PDA +10 mM As(V)). The plates were incubated at room temperature until bacteria became visible and could be isolated. The emerging bacteria were isolated and maintained in LB broth (Sigma-Aldrich) at room temperature. All bacteria were sensitive to streptomycin (100 g L^−1^)

### 2.6. Molecular Identification and Characterization of Endophytic Bacteria

All bacteria were identified by 16S rDNA sequencing according to Molina et al. [47]. Total genomic DNA was extracted by use of a DNA extraction kit (MoBio Laboratories, Solano Beach, CA, USA) and the 16s rDNA sequence was amplified using universal primers 16sF (5′-AGAGTTTGATCCTGGCTCAG-3′) and 16sR (5′-CTACGGCTACCTTGTTACGA-3′). The PCR products were purified using a PCR purification kit (Qiagen) and sent to Genewiz Inc. (South Plainfield, NJ, USA) for sequencing. The sequences were BLAST searched on the NCBI GenBank database to find the closest matches.

Bacterial As(V) Minimum Inhibitory Concentrations (AMIC) was determined as the lowest concentrations of arsenate that inhibit visible growth of the isolates. To test the AMIC, bacteria were grown in LB or PDA agar plates with concentrations from 125 μM to 450 μM. Bacteria were grown on LB plates supplemented with 10 mM As(V). After 10 days, bacteria were harvested and washed with sterile water, three times, at 13,000 rpm for 5 min each. The last pellet was resuspended in 1 mL of sterile and deionized water and used to determinate total arsenic and their species according to García-Salgado et al. [48]. Speciation studies were performed by high performance liquid chromatography-photo-oxidation-hydride generation-atomic fluorescence spectrometry (HPLC-(UV)-HG-AFS), using both anion and cation exchange chromatography; while total arsenic concentrations were determined by inductively coupled plasma atomic emission spectrometry (ICP-AES). 

Production of indole acetic acid (IAA) by bacteria in broth cultures was assessed by the colorimetric method [49] using Salkowski reagent [44]. The test to evaluate inhibition of potentially pathogenic fungi by endophytic bacteria was done on LB plates using the dual culture technique according to Verma et al. [44]. *Alternaria* sp., a potential pathogen [50,51] isolated from the surface of the *Jasione* seeds, was used as a test fungus. Finally, bacteria were screened for phosphate solubilization by a plate assay method using Pikovskaya agar media [52]. Those bacteria with a clear zone around colonies were considered to be phosphate solubilizers. 

### 2.7. Inoculation of Single Bacteria and Bacterial Mixture onto Gnotobiotic Seeds

To test the effects of the horizontal transfer of bacteria on the response of the gnotobiotic seedling to As stress, bacteria were added to the gnotobiotic seeds of *J. montana*. The gnotobiotic plant was achieved by washing the hypochlorite treatment and T2. 

One treatment was achieved by not adding bacteria to the gnotobiotic seeds, another treatment was adding *Pantoea conspicua* MC-K1 (the best PGPB and As resistant bacterium), adding *Arthrobacter* sp. MC-D3A (non-helper and non-As resistant bacterium), and the last one treatment was adding an artificial mixture prepared with *Pantoea conspicua* MC-K1, *Kocuria rosea* MC-D2, *Kocuria* sp. MC-K2, *Rodococcus rhodochorus* MC-D1 and *Arthrobacter* sp. MC-D3A. The inoculum was prepared by adding bacteria to a glass flask with 50 mL of LB and incubating at laboratory ambient temperature in a continuous orbital shaker for 5 days. After that, 1 mL of each culture (single bacterium or mixture culture) was centrifuged at 2000 rpm for 5 min. Pellets were suspended in phosphate-buffer to obtain aliquots of each bacterium and the mixture (5 bacteria together) with a cell density value of 700, at 600 nm in a spectrophotometer (Spectronic Genesys; Thermo Electron Corp.). A total of 5 μL of each single bacterium or bacterial mixture was inoculated on the gnotobiotic seedlings which were kept at room temperature under fluorescent lights for 15 days. We analyzed several physiological parameters, including germination, hypocotyls, lateral hairs, cotyledons and healthy seedling percentage. 

Each treatment was replicated 5 times and each plate contained 30 seeds planted on 0.7% agarose media with 125 µM As(V) in media (or control) and inoculated (or not) with two single bacteria or bacterial mixture. 

A total of 10 treatments; inoculum effect (×5), arsenic effect (×2). Each treatment was replicated 5 times and 30 seeds were plating on 0.7% agarose media. Those treatments with As in agarose contained 125 µM of As(V). 

### 2.8. Statistical Analysis

All data were analyzed using R software. Germination, development of roots, and healthy plants were compared between different treatments with GLMA binomial function was used because parameters were binomial. 

## 3. Results

### 3.1. Molecular Identification and Characterization of Bacteria

The identification of endophytic bacteria from plants collected from arsenic contaminated soil is shown in Table 1. They belong to Gamma-proteobacteria (*Pantoea conspicua*) and Actinobacteria (*Kocuria rosea*, *Kocuria* sp. *Rodococcus rhodochorus* and *Arthrobacter* sp.). In general (except for *Arthrobacter* sp.), they behave as plant growth-promoting bacteria (PGPB) with qualities such as, production of growth hormone (IAA), inhibition of potentially pathogenic fungi, *Alternaria* sp., or resistance to high arsenic concentrations (Table 1). All bacteria analyzed behave as As(V) transformer bacteria into organic arsenic species, with biotransformation rates higher than 30%. In the case of *Kocuria* strains, free As(III) was not detected. Moreover, *P. conspicua* was a phosphate solubilizer. The bacterial mixture did not solubilize inorganic phosphate but retained other plant helper qualities. 

### 3.2. Anatomical and Physiological Effects of Arsenic on Seedlings

When seedlings were germinated on media with different arsenate concentrations (Figure 2) morphological changes were observed. Germination was significantly affected by arsenic concentration above 500 µM As(V). The percentage of well-developed roots and lateral hair abundance also decreased, with arsenic concentration being drastically decreased with arsenic concentrations above 125 µM As(V). A total of 1 mM arsenate concentration was thus used to evaluate anatomical changes provoked by arsenic. Because of this preliminary screening, 125 µM arsenate concentration was selected for inoculation experiments on the gnotobiont, given that germination, root system and hypocotyl development was not affected but parameters such as cotyledon production and plant viability were variable and easily measurable parameters. Seedlings of *J. montana* not treated with As(V) showed development of well-developed root systems with abundant lateral hairs (Figure 3A) and healthy cotyledons with oxidation reactions only in the vascular system (Figure 3C). A matrix was observed around them, occupied by aniline blue stained epiphytic bacteria (Figure 3A,B). In addition, brown-stained intracellular bacteria exhibiting elevated H_2_O_2_ presence were also observed in root cells (Figure 3B). Coccoidal bacteria were visualized in apoplastic spaces of root cells (Figure 3D) and bacterial rods were frequently seen within parenchyma cells (Figure 3E). Seedling plants watered with 1 mM arsenate solution showed anatomical changes with strong signs of oxidative stress in the cotyledons, reduction of chlorophylls (Figure 4C) and incipient tumors in the root (Figure 4D). In addition, frequently collapsed root cells were observed (Figure 4A). However, coccoidal bacteria (Figure 4A) and bacterial rods within parenchyma cells (Figure 4B) were detected in control seedlings (Figure 3).

### 3.3. Anatomical and Physiological Effects on Gnotobiotic Jasione (1 mM Arsenate)

The effect of the [As], removing the epiphytic bacteria and the antibiotic treatments interaction on germination was not significantly different (Table 2). However, the interaction between treatments with [As] and treatment with hypochlorite were significantly difference (Table 2; Figure 5A,B). Regardless of the elimination of epiphytic bacteria with hypochlorite solution, the germination rate was significantly lower in all sterilization treatments (T2, T3, T4), compared to the control (T1), this reduction was more drastic under arsenic conditions. The lowest germination rate was always in T4, when the antibiotic was added before and after planting (Figure 5A,B). The treatment T4 was so strong that seedlings did not show any effect of the As (Figure 5B). Removal epiphytic bacteria (washing seeds in 2% NaClO), significantly improved germination on 1 mM As(V) agarose (Figure 5A). When we tested the success of removal of epiphytic bacteria in the seed plated onto different bacterial culture media, no bacterial presence was observed in any of the plates (data not shown).

The capacity of *J. montana* to generate hypocotyls was reduced with any antibiotic treatment (T2, T3 and T4) and was significantly reduced when the sterilization treatment became more drastic, a phenomenon that was independent of the presence or absence of arsenic in the media and all other factors (Figure 5C). In those seeds germinated in the absence of arsenic, the preliminary washing with sodium hypochlorite did not affect their ability to generate hypocotyls; however, when the seeds were subjected to As stress, the elimination of epiphytes favored the development of hypocotyls (Figure 5C). 

The visualization, under an optical microscope, of the development of the seeds in one of the sterilization treatments (light treatment, T2), after germinating with non-stressful conditions, meristematic cells showed deterioration and roots were collapsed (Figure 6A). In this treatment, we visualized many lateral root hair primordia that did not elongate (Figure 6D). Frequently, the embryos appeared contorted (Figure 6F). Fewer bacteria on the meristem cell wall or endophytes in parenchyma cells were observed (Figure 6B,C). 

### 3.4. Seedling Growth Promotion Experiments with Single Bacterium or Bacterial Mixture (125 μM Arsenate)

When holobiotic seedlings were developed on arsenic culture media (125 μM As(V)), the percentage of germination, development of hypocotyls, roots, lateral hairs and healthy plants were not affected (Figure 2). However, in gnotobiotic-plants, the percentage of healthy plants was determined by the two factors: intact biome and stress by arsenic (Figure 7, Table 3), causing both parameters, a statistically significant reduction in the number of healthy plants. The inoculation of a single PGPB (*P. conspicua*-MCK1) shows a tendency for the recovery of the plant, both in arsenic-enriched and control media, while bioaugmentation with *Arthrobacter* sp. does not help in the recovery of the plants either in stress situation and control. Inoculation with the bacterial mixture (containing all five endophytic bacteria) allows a clear recovery of the plants in control conditions. However, the interactions generated in the consortium under stress condition, probably as a consequence of *Arthrobacter* sp (non-helper and non-As resistant bacteria), prevent the recovery of plants.

## 4. Discussion

Our results have shown how *J. montana* can support 125 μM arsenate concentration without significant changes to germination, hypocotyls and root development, although only 50% of the seeds developed healthy plants. This threshold concentration has also been described in other As-tolerant species, such as *Betula celtiberica* [21]. The percent of germination and root development were drastically reduced at 500 and 250 μM As(V), respectively. These physiological parameters are sensitive to arsenic pollution because early roots are the first contact points with this toxic compound [13,53]. Liu et al. [54] suggest that some of the defense mechanisms have not yet developed in this early stage of seedling development. Harminder et al. [55] conclude that arsenic causes a reduction in root elongation by inducing oxidative stress, which is related to enhanced lipid peroxidation, but not to H_2_O_2_ accumulation. In agreement with previous works [5,53,56] our results confirm the hypothesis that plants stressed with arsenate (1 mM As(V)) suffer free ROS that can be detoxified because of the activation of superoxide dismutase (with an increase in the production of H_2_O_2_). Coleoptiles showed cellular oxidative damage with chlorotic appearance. Reduction in chlorophyll content because of arsenic has previously been described [6] and has been related to decrease in leaf gas-exchange [57] and photosynthesis damage [58]. The presence of root tumors can also be an effect of the oxidative damage caused to the cells [59]. Broken cells in the outer layer cortex as an arsenic consequence has been previously observed in other plants [8]. Endophytes and epiphytic bacteria in root and shoot tissues, which are seemingly healthy, points to a possible implication of microbiome in the acclimatization to this stressful situation.

The sterilization methods to achieve a gnotobiotic *Jasione* showed that the washing of the seeds with 2% NaClO did not completely remove the bacteria (Figure 6E,F), but those that were observed microscopically after treatment, either were dead or were non-cultivable bacteria, because they did not emerge when plated on suitable media for bacterial growth. This sterilization protocol resulted in an increase in germination and early development, which may be a consequence of the reduction of phytopathogenic microorganisms associated with the seed surface [60,61]. The treatment of the seeds with streptomycin, during the 24 h prior to being plated on germination media (Figure 5), resulted in a significant reduction in the early development of seedlings [44]. Verma et al. [44] demonstrated how plant–gnotobiont model achieves, given that the plant tissues stay free of bacteria. 

The most drastic treatment of sterilization using streptomycin throughout the development period resulted in a greater reduction in germination and total annulation of further development. These results could be explained because of the pernicious effect of the antibiotic on organelles, such as mitochondria [62], that would result in irreversible damage to seedling development. Minder et al. [63] describes how antibiotics in agricultural soils damage plants and soil microbial communities. However, other possible explanations could be that microbiota is intimately involved in the development of seedlings [44,64,65,66] and antibiotics, by decreasing the internal microbial potential, lead to significant damage to plants. In fact, inoculation experiments on the gnobiotic system confirm that indigenous endophytic bacteria from J. *montana* play important roles in modulating seedling development [44], especially under stress conditions. In fact, the inoculation with only one PGP microbe, such as strain *P. conspicua* MC-K1, allowed the partial recovery of development, both in the presence and in the absence of arsenic. *P. conspicua* MC-K_1_ has an ability to protect the host plant from arsenic stress, apparently through several plant growth-promoting mechanisms. It is able to synthesize IAA that activates root growth, nodule production and development of lateral hairs [65]. It also has antifungal capacity that can defend the seeds in these early stages of development from attack of pathogenic fungi. In fact, *P. agglomerans* (genetically close to *P. conspicua*) had antagonistic properties towards *Alternaria* sp. [67] and is a diazotrophic bacterium [68]. *P. conspicua* MC-K_1_ also behaves as an inorganic phosphate solubilizer. Elevated levels of arsenic in soils also interfere with P uptake, but bacteria can compensate for this deficiency by solubilizing P, which may be particularly important in the early stages of seedling development [69,70]. Nevertheless, the most important helping qualities are a higher As(V) MIC (450 mM) and the capacity to accumulate As and reduce As(V) to As(III), metabolizing the latter towards less toxic arsenic organic forms (Table 1). Arsenic detoxification mechanisms reported include either methylation and subsequent transformation into volatile forms [25], or strong binding of As(III) to –SH groups of cytosolic proteins [71]. We did not detect methylation forms, but it is evident that complex organic forms (perhaps protein complexes, phytochelatins, etc.) were produced and these probably had reduced toxicity. Other *Pantoea* strains have also been shown to have high As(V) MIC [72]. Wu et al. [73] reported *Pantoea* sp. as an As(V) reducer for the first time and, recently, Tian and Jing [74] sequenced its genome to characterize the molecular mechanism of arsenate reduction. Given that *P. conspicua* is a PGP bacterium then transforms As(V) into As(III) and mainly into organic forms, and also favors the growth of *J. montana* under arsenic stress, it is possible that it is involved in the resistance of *J. montana* to arsenic stress. 

The behavior of *Arthrobacter* strains against arsenic is variable. Some strains are able to survive at high concentrations [75] but others only support low concentrations [76], like *Arthrobacter* sp. MC-D_3_A. *Arthrobacter* species are widely present in soils and environmental polluted with chemicals and heavy metals [75] and in water As contaminated [77]. It has been described as an endophytic PGP bacterium [78], but some strains have been recognized as opportunistic pathogens [79]. *Arthrobacter* MC-D_3_A does not behave like a PGPB and is not tolerant to As concentrations higher than 10 mM, at least, in aposymbiotic conditions. Use of a complex inoculum, rather than single strains, improved arsenic sequestration efficiency of hyper-accumulator plants [22,31]. Our bacterial mixture behaved as a helper inoculum in control conditions but did not show positive effects under arsenic stress, even though four of the five species had plant growth promotional features. Bacterial mixture can evolve very rapidly [80], generating antagonism or competitive interactions between species and causing community collapse and subsequent loss of plant protection [81]. The presence of *Arthrobacter* sp. MC-D_3_A in the bacterial mixture may be the main reason that this complex inoculum did not help *J. montana* in physiological recovery under stress conditions since this bacterium appeared to negatively affect seedlings. 

The plant-metaorganism (or plant-holobiont) bio-components show different contributions against arsenic stress. The seedling by itself is able to modulate the pernicious effect of 125 mM arsenic in early stages of development (germination, hypocotyl and lateral hair production) but in absence of the microbiome the ability to develop healthy plants is drastically reduced. The inoculation with a *P. conspicua* MC-K_1_ in gnotobiotic seedlings seems to improve arsenic acclimatization capacity. Possibly the interaction of this microbe, together with other components of the microbiome, and the plant itself, are responsible for the survival of the metaorganism under arsenic stress conditions. Since *P. conspicua* was isolated from adult plants but beneficial in early germination, it is possible that seedings recruit it from the soil in the early stages of development (horizontal transfer) or it may be vectored with the seed (vertical transfer). Additional research is needed to evaluate these hypotheses. *Arthrobacter* strains can have negative effects for the plant under stressful conditions. The relationship of this bacterium to others through competition–facilitator interactions can also drive the response of the plant to these conditions. Therefore, the tradeoff between helper and non-helper bacteria, together with the plant, would determine the plant-metaorganism response to As contamination [82].

## 5. Conclusions

The physiological damage in plants is greater the more drastic the sterilization process is. When we use light sterilization conditions, *J. montana* seeds without your streptomycin-sensitive microbiome may germinate but they show significantly reduced ability for seedling development, especially under As stress. The type of bacteria (helper or nonhelper) to the gnotobiotic plants shows different responses, from a tendency to recovery of development to a reduction in this capacity. The interactions between bacteria determine the response of the plant that varies significantly in environments contaminated by arsenic. 

## Figures and Tables

**Figure 1 microorganisms-09-00045-f001:**
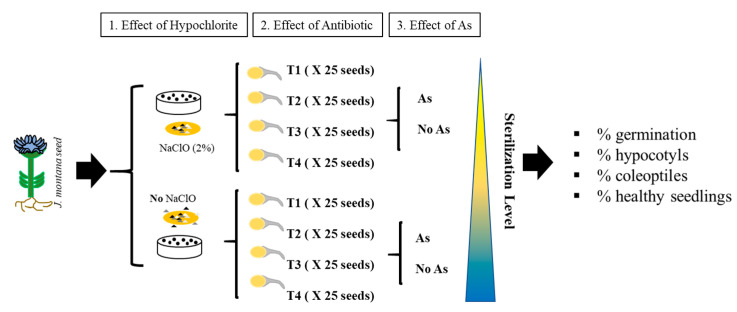
(1) Effect of hypochlorite with 2 levels (surface sterilization to eliminate epiphytes). (2) Effect of antibiotic with 4 levels. Seeds with no antibiotic (T1), seeds with antibiotic 24 h before germination (T2), seeds with antibiotic during germination (T3), seeds with antibiotic before and during germination (T4 = T2 + T3). Levels of sterilization (non, light, severe and extreme). The antibiotic was streptomycin for the elimination of endophytes. (3) Effect of As with 2 levels. A total 25 seeds in each treatment (16 treatments). (3) Germination and development was carried out on 0.7% agarose plates.

**Figure 2 microorganisms-09-00045-f002:**
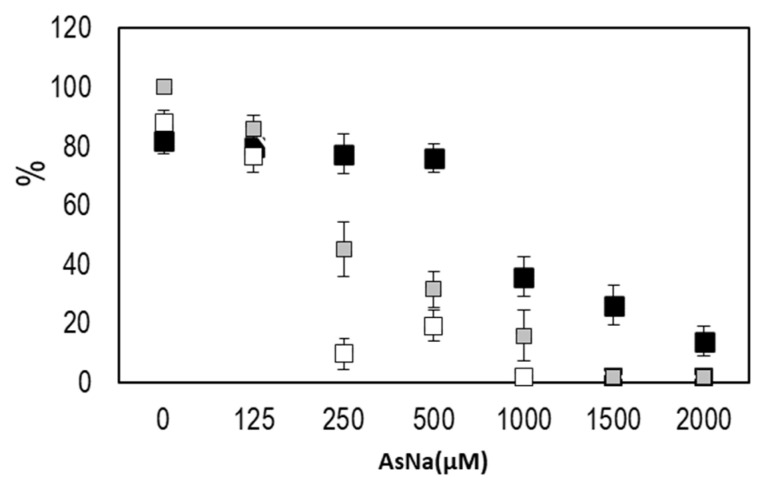
*J. montana* planted on different concentration of arsenate. Percentage of germination (black squares), percentage of seedling plants that develop roots (gray squares) and percentage of seedling plants with lateral hairs (white squares).

**Figure 3 microorganisms-09-00045-f003:**
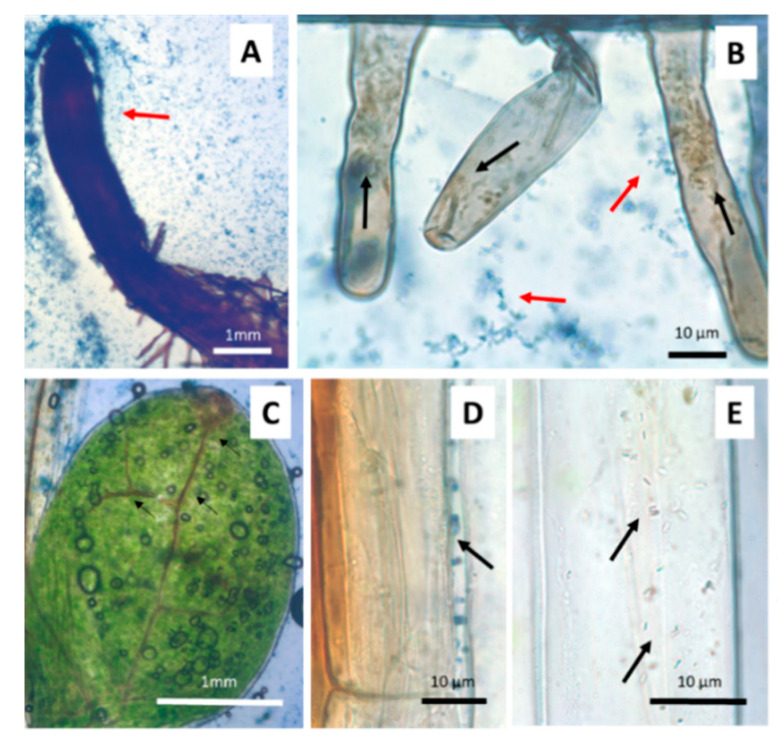
*J. montana* seedling plant control. (**A**). Root surrounded by a bacterium community (red arrow). (**B**). Detail of the matrix that surrounds the root hairs where aniline blue stained bacteria can visualize (red arrow). Brown bacteria stained with DAB/peroxidase for reactive oxygen (black arrows) within lateral hairs. (**C**). Healthy cotyledons without signs of deterioration due to oxidative stress. Defection of H_2_O_2_ accumulation in circulatory system (black arrows). (**D**). Aniline blue coccoidal bacteria in apoplastic space of root (black arrow). (**E**). Bacteria rods stained with aniline blue inside hypocotylous parenchyma (black arrows).

**Figure 4 microorganisms-09-00045-f004:**
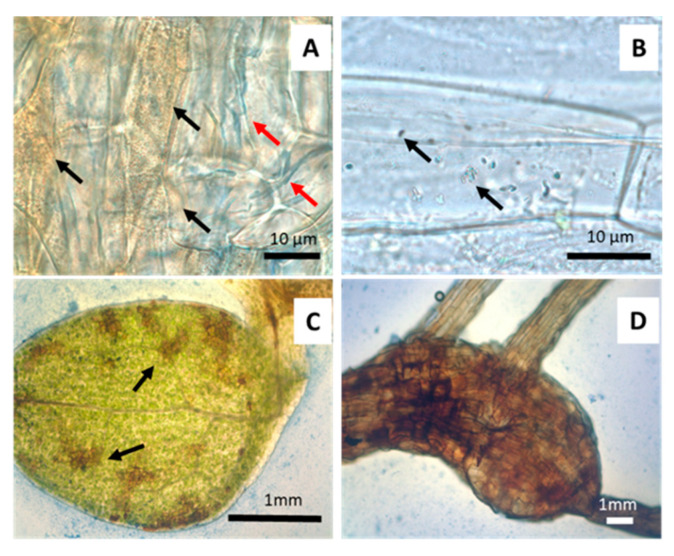
*J. montana* seedling plants germinated on agarose plates and watered with 1 mM As(V). (**A**). Coccoidal bacteria staining brown on cell wall surface in some root cells (black arrows) with meristemic cells damaged (red arrow). (**B**). Bacteria rods (arrows) with capacity to stain with aniline blue (arrows) inside hypocotylous parenchyma. (**C**). Oxidative deterioration (arrows) in cotyledons with H_2_O_2_ accumulation. (**D**). Incipient root tumor caused by arsenic oxidative stress.

**Figure 5 microorganisms-09-00045-f005:**
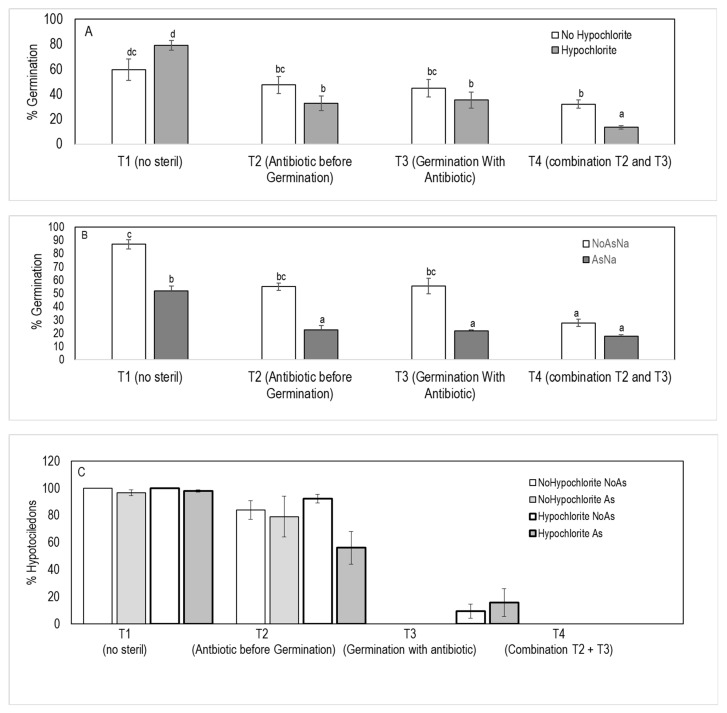
Experiment to analyze the effect of arsenic and hypochlorite washes on functional gnotobiotic *J. montana* obtained by different levels of sterilization. T1, no sterilization; T2, washing the seeds with streptomycin 24 h before planting; T3, germination for ten days in streptomycin media; T4, combination of T2 and T3. (**A**) and (**B**) show the effect of interaction treatment with As and treatment with hypochlorite on germination. (**C**), the percentage of hypocotyls developed. Different letters mean statistically significant differences.

**Figure 6 microorganisms-09-00045-f006:**
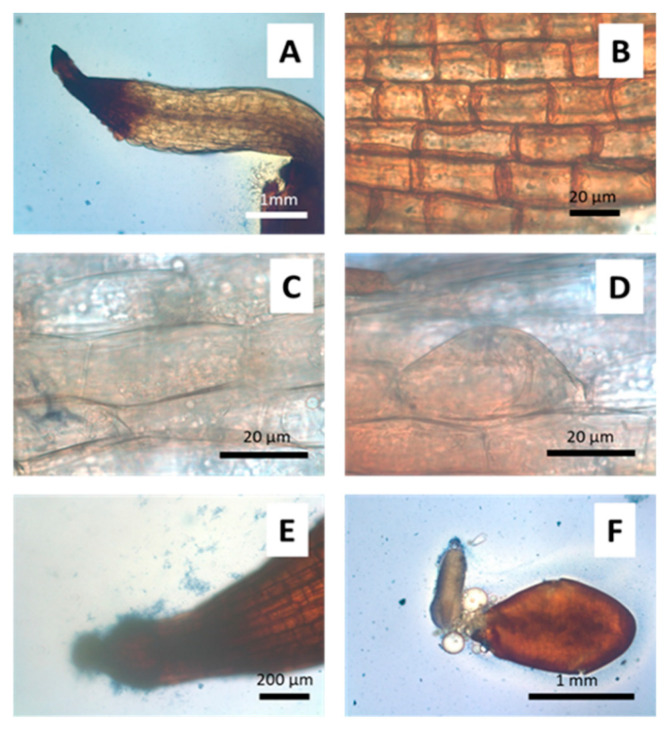
Gnotibiotic *J. montana* (endophytic and epiphytic bacteria removal) without As. (**A**). Root collapsed and intensely stained with DAB/peroxidase for reactive oxygen. Root (**B**) and shoot (**C**) tissue without bacteria. (**D**). Early and non-developed root hair initial. (**E**). Dead or non-cultivable bacteria around the root appendix. (**F**). Embryo aborted before development.

**Figure 7 microorganisms-09-00045-f007:**
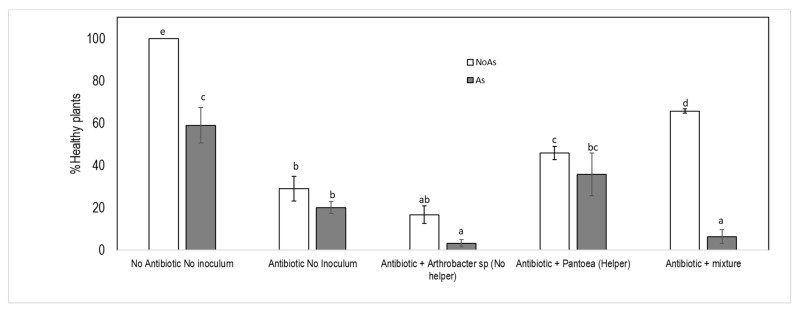
Gnotobiotic *J.montana* response to As (125 uM As(V)) under different treatments inoculated with helper bacteria, a non-helper bacterium or a mixture of bacteria. Natural control plants, gnotobiotic plants treated with streptomycin 24 h before planting and not inoculated, inoculated with *Artlrobacter* sp (non-helper bacteria), inoculated with a *Pantoea conspicua* (helper bacteria) and inoculated with a mixture of endophytic bacteria (*Pantoeaconspicua, Kocuriarosea, Kocuria sp.Rodococcus rhodochorus,Arthrobacter* sp.).

**Table 1 microorganisms-09-00045-t001:** Molecular identification (similarity > 99.5%) and characterization of five endophytic bacteria (and a mixture of all of them) from *J. montana* growing on arsenic contaminated soil: ability to solubilize inorganic phosphate, to produce IAA, to inhibit the growth of *Alternaria* sp., and arsenic species and total arsenic concentrations within bacteria growing at 10 mM arsenate.

	Molecular Approach *	P	IAA Production	% Fugal Inh.	AMIC	As(V) ^a^	As(III) ^a^	Total As ^b^	As(V) Biot. ^c^
Strains			(μg mL^−1^ DO^−1^)	LB	mM	mgL^−1^	mgL^−1^	mgL^−1^	%
MC-K1	*P.conspicua* ^T^	+	1.96 ± 0.07	40	450	0.125 ± 0.3 10^−3^	0.091 ± 0.001	0.63 ± 0.03	66
MC-K2	*Kocuria* sp.	−	0.46 ± 0.10	53	450	0.561 ± 0.008	n.d.	0.88 ± 0.02	36
MC-D1	*R. rhodochorus* ^T^	−	0.76 ± 0.10	40	450	0.266 ± 0.005	0.022 ± 0.001	0.67 ± 0.03	57
MC-D2	*K. rosea* ^T^	−	1.05 ± 0.07	46.7	450	0.152 ± 0.001	n.d.	0.54 ± 0.04	72
MC-D3A	Arthrobacter sp.	−	n.d.	73	10	n.a.	n.a.	n.a.	
Mixture bacteria		−	4.46 ± 0.01	20	200	n.a.	n.a.	n.a.	

LB: *Luria Bertani medium*, n.d.: non-detected, n.a.: non-analyzed, As(V)MIC: As(V) minimal inhibition concentration. ^a^ Determined by anion exchange HPLC-(UV)-HG-AFS. ^b^ Determined by ICP-AES. ^c^ Estimated as the difference, expressed as a percentage, between total As and the sum of As species found in samples. * Whenever possible, molecular identification was carried out with the type strain ^T^.

**Table 2 microorganisms-09-00045-t002:** GLM analyses to test the effect of arsenic and hypochlorite washes on germination process of functional gnotobiotic *J. montana* obtained by different levels of sterilization (T1, T1, T3 and T4). Degrees of freedom (df), Deviance Residual (DR) and the probability associated to the estimation (Pr). The zero concentration of arsenate is the value of reference.

Parameter	Factor	df	DR	Pr (>/Chi/)
Germination	Treatment	3	365.08	<0.001
	Hypochloryte	1	16.05	<0.001
	[As]	1	258.75	<0.001
	Treatment:Hypochloryte	3	84.21	<0.001
	Treatment:As	3	21.61	<0.001
	Hypochloryte:As	1	3.07	ns
	Treatment:Hypochloryte:As	3	6.77	ns
Hypocotiledons	Treatment	3	1379.28	<0.001
	Hypochloryte	1	3.83	ns
	[As]	1	225.45	<0.001
	Treatment:Hypochloryte	3	46.84	<0.001
	Treatment:As	3	4.49	ns
	Hypochloryte:As	1	0.11	ns
	Treatment:Hypochloryte:As	3	11.01	0.011

**Table 3 microorganisms-09-00045-t003:** GLM analyses to test the effect of arsenic (125uM As[V]) on germination and healthy development of *J. montcma* obtained by different treatments. Degrees of freedom (df), Deviance Residual (DR) and the probability associated to the estimation (Pr). Natural control plants (T1), gnotobiotic plants treated with streptomycin 24 hours before planting and no inoculated (T2), inoculated with *Arthrobacter* sp (non-helper bacteria; T3), inoculated with a *Pantoea*. *conspicua* (helper bacteria; T4) andinoculated with a mixture of bacteria (T5).

Parameter	Factor	df	DR	Pr(>/Chi/)
Germination	Treatment	4	7.33	0.11
	[As]	1	0.074	0.78
	Treatment: [As]	4	2.661	0.61
Healthy plants	Treatment	4	170.315	<0.001
	[As]	1	46.072	<0.001
	Treatment: [As]	4	27.251	<0.001

## Data Availability

MDPI Research Data Policies” at https://www.mdpi.com/ethics.

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
