# Peer review of "A Gnotobiotic Model to Examine Plant and Microbiome Contributions to Survival under Arsenic Stress"

_microorganisms, 2020, doi:10.3390/microorganisms9010045_

Round 1

Reviewer 1 Report

The manuscript entitled "A gnotobiotic model to examine plant and microbiome contributions to survival under arsenic stress" describes some interesting interactions between plant and microorganisms during germination of seed in condition of arsenic stress. Jasione montana seeds were used as model. Authors obtained few isolates of endophytic bacteria from first set of seeds and then they applied these bacteria to another batch of sterilized seeds both with and without arsenic stress. They used several options for sterilization, including sodium hypochloride and streptomycin treatment. There was observed significant negative effect of streptomycine treatment emphasized by presence of single bacterium while presence of multiple bacteria recovered % of healthy plants without presence of arsenic stress. Application of plant probiotic bacteria Pantoea conspicua increased % of healthy plants even in presence of Arsenic stress. Authors used ROS depended staining and microscopy for evaluation of plant health under arsenic stress and bacterial inoculation. Hypothesis is widely stated however only few microorganisms was evaluated.

First of all. Manuscript is very poor written, It has a lot of typos and some wrong stylization. Changing from I to Y and vie-versa in very common through whole article. It also contains hard to understand passages for example L50 role of bioma plant, L411 plants whose genome is removed. Thus article needs significant language correction.

Research design seems little complicated for me (or not written to be easy to read).

In table1, the column "molecular identification" should be renamed to the "best hit" or something like because it looks like type strain was used in study

In figure2 connection lines should not be used as it implicate that X axis is a scale (linear, exponential, or logarithmic), but it is not.

Luria Bertani - https://asm.org/getattachment/5d82aa34-b514-4d85-8af3-aeabe6402874/LB-Luria-Agar-protocol-3031.pdf

Author Response

Referee 2

The manuscript entitled "A gnotobiotic model to examine plant and microbiome contributions to survival under arsenic stress" describes some interesting interactions between plant and microorganisms during germination of seed in condition of arsenic stress. Jasione montana seeds were used as model. Authors obtained few isolates of endophytic bacteria from first set of seeds and then they applied these bacteria to another batch of sterilized seeds both with and without arsenic stress. They used several options for sterilization, including sodium hypochloride and streptomycin treatment. There was observed significant negative effect of streptomycine treatment emphasized by presence of single bacterium while presence of multiple bacteria recovered % of healthy plants without presence of arsenic stress. Application of plant probiotic bacteria Pantoea conspicua increased % of healthy plants even in presence of Arsenic stress. Authors used ROS depended staining and microscopy for evaluation of plant health under arsenic stress and bacterial inoculation. Hypothesis is widely stated however only few microorganisms was evaluated.

Although the hypothesis of the importance of the biome in the adaptive capacity of plants is widely documented, there are only about 30 works in which the importance of the biome in the development of the plant under conditions of arsenic stress is raised. In none of them have functional gnotibionts been used to establish the relative importance of plants and the microbiome.

On the other hand, we decided to work with a single bacterium to establish a cause-effect relationship in the simplest possible way. We ourselves have shown that the complexity of the inoculum (mixture bacteria) can significantly alter the results, as suggested by Thijs et al., 2016 referring to the effect of the inoculum on global interactions between microorganisms.

First of all. Manuscript is very poor written, It has a lot of typos and some wrong stylization. Changing from I to Y and vie-versa in very common through whole article. It also contains hard to understand passages for example L50 role of bioma plant, L411 plants whose genome is removed. Thus article needs significant language correction.

Thank you for your patience. We apologize for any linguistic errors in the text. English synthesis and semantic of the manuscript, have been revised and improved by Dr. James White.

Research design seems little complicated for me (or not written to be easy to read).

We have included a new figure to facilitate your understanding

In table1, the column "molecular identification" should be renamed to the "best hit" or something like because it looks like type strain was used in study

You are right. We have clarified that in the table.

In figure2 connection lines should not be used as it implicate that X axis is a scale (linear, exponential, or logarithmic), but it is not.

Thank you. The figure has been modified

Luria Bertani - https://asm.org/getattachment/5d82aa34-b514-4d85-8af3-aeabe6402874/LB-Luria-Agar-protocol-3031.pdf

We prepare all media according to the manufacturer's instructions

Reviewer 2 Report

The article by Molina et al aims to achieve an interesting goal and if successful which could serve for functional and fundamental studies in plant-bacteria interactions. However, the objective is not entirety achieved and therefore, the results obtained must be considered with caution. Nevertheless, it is necessary to start to pave the way in this field and in this sense, this work brings some interesting aspects and results, such as effects of arsenic on seedlings and the reduction of seedlings microbiome have on germination rate even under arsenic conditions.

However, the article has serious gaps in the article that should be considered:

  • The authors claim that they obtained a gnotobiotic plant model system, but in fact this is not proven since the observation of bacteria (dead or non-cultivable) were observed in the different treatments.

The authors refer to the methodology used in this work was similar to Verma et al (2017), but do not discuss the possibility of different plants present different sensitivity to antibiotics. In fact, there is a very interesting work that reveals that “even comparatively small concentrations of antibiotics as typically found in the soil of agricultural landscapes can delay the time of germination and differently affect trait development of different plant species, with effects depending on species, traits, antibiotics and concentrations. (Minden, V., Deloy, A., Volkert, A. M., Leonhardt, S. D., & Pufal, G. (2017). Antibiotics impact plant traits, even at small concentrations. AoB PLANTS9(2), plx010. https://doi.org/10.1093/aobpla/plx010). This raised the question: Is it the antibiotic that eliminates epiphytes and endophytes or is it the antibiotic that harms plant tissues? The authors discuss very lightly of this possibility in the discussion. This could be further discussed. Why did the authors not test different antibiotics?

Why the bacterial strains have not been tested at different concentrations of antibiotics?

Why the authors designate Arthrobacter sp. as non-helper bacteria? Just because this strain didn’t promote seedlings growth / germination…?

Figures are incorrectly positioned throughout the text. Note that the figures must appear after mentioned in the text.

The numbering of the references in the list needs to be revised (it is numbered in duplicate).

There is some hurry in writing the article. Unfinished sentences or punctuation are observed.

Author Response

The article by Molina et al aims to achieve an interesting goal and if successful which could serve for functional and fundamental studies in plant-bacteria interactions. However, the objective is not entirety achieved and therefore, the results obtained must be considered with caution. Nevertheless, it is necessary to start to pave the way in this field and in this sense, this work brings some interesting aspects and results, such as effects of arsenic on seedlings and the reduction of seedlings microbiome have on germination rate even under arsenic conditions.

However, the article has serious gaps in the article that should be considered:

The authors claim that they obtained a gnotobiotic plant model system, but in fact this is not proven since the observation of bacteria (dead or non-cultivable) were observed in the different treatments.

In some treatments, bacteria were observed on the surface under the microscope, but if they are dead, we have achieved our objective. Obtaining a completely sterile plant is certainly practically impossible. What we want is to achieve a "functional gnotobiont", which allows us to demonstrate what happens when the plant loses part of its microbiome and how it recovers, at least partially, when even a single bacterium is inoculated.

We clarified this is a  partial gnotobiont in line 74:

“In order to meet this goal, we use a partial gnotobiotic functional (without streptomycin-sensitive biome) of J. montana to evaluate the physiological response of J. montana under As stress conditions and to analyze the relative roles of the plant and the associated microbiome in resistance to As stress”.

The authors refer to the methodology used in this work was similar to Verma et al (2017), but do not discuss the possibility of different plants present different sensitivity to antibiotics. In fact, there is a very interesting work that reveals that “even comparatively small concentrations of antibiotics as typically found in the soil of agricultural landscapes can delay the time of germination and differently affect trait development of different plant species, with effects depending on species, traits, antibiotics and concentrations. (Minden, V., Deloy, A., Volkert, A. M., Leonhardt, S. D., & Pufal, G. (2017). Antibiotics impact plant traits, even at small concentrations. AoB PLANTS9(2), plx010. https://doi.org/10.1093/aobpla/plx010). This raised the question: Is it the antibiotic that eliminates epiphytes and endophytes or is it the antibiotic that harms plant tissues? The authors discuss very lightly of this possibility in the discussion. This could be further discussed.

Thank you for this bibliographic reference that we did not have. In the discussion we mentioned this possibility (Line 306). Indeed, the antibiotic can damage mitochondria and chloroplasts, but what we show is that, even if this is happening, inoculation with Pantoea MC-12 significantly improves the number of plants recovered under stress conditions and in non-stressful conditions. That is, at least one bacterium (or the mixture under control conditions) is involved in the development and adaptation of the plant. This is the novelty, and this is what has not been demonstrated before because whenever an inoculation was made, it was done in a holobiont plant, with its complete biome.

On the other hand, we find it very striking that, in the work by Minden et al., 2017, the damage caused by antibiotics to plants and possible damage to soil bacteria is considered, however, it is not considered that perhaps antibiotics, as we are demonstrating, also affect the plant biome. This is a real condition that is not being taken into account in environmental or agricultural decision-making.

We have improved this part of the discussion and we included this new paragraph in line 306:

Minder et al. [63] describes how antibiotics in agricultural soils damage plants and soil microbial communities. However, other possible explanations could be that microbiota is intimately involved in the development of seedlings [44,64,65,66] and antibiotics, by decreasing the internal microbial potential, lead to significant damage to plants.

Why did the authors not test different antibiotics?

Our objective is not to test different antibiotics, but simply to reduce a large number of bacteria in the plant biome (all those sensitive to streptomycin) and to see to what extent the incorporation of one of them improves the response to arsenic stress. In this way we ensure that, at least, this bacterium represents an adaptive advantage for plants under arsenic stress. We have also verified that the isolated bacteria with which we have worked are all sensitive to streptomycin. Therefore, in the gnotobiont streptomycin + bacteria were eliminated and only incorporated one of this to control the effect. We believe that we had not explained well that our bacteria are sensitive to streptomycin. We have included this information in the text

Why the bacterial strains have not been tested at different concentrations of antibiotics?

We are not interested in working at different concentrations of antibiotic, only to make sure that the protocol allows sufficient sterilization to confirm the working hypothesis. It seems to us that with this concentration we meet these two objectives: sterilize the plant with streptomycin, and therefore eliminate bacteria sensitive to this antibiotic and then inoculate with an arsenic-tolerant bacterium, which we are sure we have eliminated, and from which we want to know its effect.

Why the authors designate Arthrobacter sp. as non-helper bacteria? Just because this strain didn’t promote seedlings growth / germination…?

You are right, by inoculating the plant with it, "it doesn't help the plant" to recover its development

Figures are incorrectly positioned throughout the text. Note that the figures must appear after mentioned in the text.

This is done. Thank you.

The numbering of the references in the list needs to be revised (it is numbered in duplicate).

This is done. Thank you

There is some hurry in writing the article. Unfinished sentences or punctuation are observed.

We apologize for these errors that we believe have been resolved.

Round 2

Reviewer 1 Report

Language issues of this manuscript were solved at least to the level which I can recognize as non-native speaker

However some artifact of hurry-up preparation are still present. 

In L170-171 OD for cell counting is stated as OD at 700 or 600 nm. Which one is correct?

L197 indolyl acetic acid IAA?

L351 dot after variability

I cannot agree with first sentence of conclusion. Lower germination is very probably connected more to antimicrobial treatment then to the absence of microorganisms solely. It needs to be corrected before manuscript can be published

Some flaws in the way how gnotobionts were prepared can bias results however study still should be considered as relevant.

Author Response

Language issues of this manuscript were solved at least to the level which I can recognize as non-native speaker

However some artifact of hurry-up preparation are still present. 

In L170-171 OD for cell counting is stated as OD at 700 or 600 nm. Which one is correct?

Perhaps we have not explained ourselves well. We have modified this sentence:

Pellets were suspended in phosphate-buffer to obtain aliquots of each bacterium and the mixture (5 bacteria together) with a cell density value of 700, at 600 nm in a spectrophotometer (Spectronic Genesys; Thermo Electron Corp). 

L197 indolyl acetic acid IAA?

Yes, you are right, sorry

L351 dot after variability

We have removed this entire paragraph at the discretion of the other referee

I cannot agree with first sentence of conclusion. Lower germination is very probably connected more to antimicrobial treatment then to the absence of microorganisms solely. It needs to be corrected before manuscript can be published

We do agree. In fact, we are saying that the plant treated with antibiotics can germinate, even under stress conditions (Figure 5), although its later development is affected. We have modified this paragraph to clarify this point.

This is the new paragraph:

The physiological damage in plants is greater the more drastic the sterilization process is. When we use light sterilization conditions, J. montana seeds without your streptomycin-sensitive microbiome may germinate but they show significantly reduced ability for seedling development, especially under As stress. The bioaugmentation of these gnotobiontic plants shows responses dependent on the inoculated microorganisms, from a tendency to recovery of development to a reduction in this capacity. The interactions between bacteria determine the response of the plant that varies significantly in environments contaminated by arsenic.

Some flaws in the way how gnotobionts were prepared can bias results however study still should be considered as relevant.

Thank you very much

Reviewer 2 Report

I thank the authors for clarifying my doubts and for including the changes in the text. I understand that in the absence of a better procedure to eliminate the microbiome of plants that what has been done is in fact a possible approximation at this time.

However there are still some details that need to be improved:

L119- "H2O2" needs to be corrected

L180- It should be subsection 2.8 Statistical analysis

Table 1- the table must be moved down, which is when the reference to it first appears

Figure 5- indicate  the meaning of the letters in the graphs  (statistical differences) in the legend and include the statistical analysis in figure 5C

Figure 7- include the statistical analysis in figure

Discussion- the indication of figures and tables in the discussion is not necessary

Conclusion- the conclusions should be a summary of the main findings of the work. It makes no sense to end the conclusions with reference to another work

Author Response

I thank the authors for clarifying my doubts and for including the changes in the text. I understand that in the absence of a better procedure to eliminate the microbiome of plants that what has been done is in fact a possible approximation at this time.

However there are still some details that need to be improved:

L119- "H2O2" needs to be corrected

Done!

L180- It should be subsection 2.8 Statistical analysis

Yes, you are right

Table 1- the table must be moved down, which is when the reference to it first appears

OK, this is done too.

Figure 5- indicate  the meaning of the letters in the graphs  (statistical differences) in the legend and include the statistical analysis in figure 5C

We included this information in figure.

Figure 7- include the statistical analysis in figure

Done!

Discussion- the indication of figures and tables in the discussion is not necessary

Ok, we have removed them

Conclusion- the conclusions should be a summary of the main findings of the work. It makes no sense to end the conclusions with reference to another work

This is true. We remove this paragraph.